# ViPER: Visibility-based Pursuit-Evasion via Reinforcement Learning

**Yizhuo Wang**[1*]   **Yuhong Cao**[1*]   **Jimmy Chiun**[1]   **Subhadeep Koley**[2]
**Mandy Pham**[3]   **Guillaume Sartoretti**[1]
[1]National University of Singapore   [2]IIEST Shibpur   [3]UC Berkeley

**Abstract:** In visibility-based pursuit-evasion tasks, a team of mobile pursuer robots with limited sensing capabilities is tasked with detecting all evaders in a multiply-connected planar environment, whose map may or may not be known to pursuers beforehand. This requires tight coordination among multiple agents to ensure that the omniscient and potentially arbitrarily fast evaders are guaranteed to be detected by the pursuers. Whereas existing methods typically rely on a relatively large team of agents to clear the environment, we propose ViPER, a neural solution that leverages a graph attention network to learn a coordinated yet distributed policy via multi-agent reinforcement learning (MARL). We experimentally demonstrate that ViPER significantly outperforms other state-of-the-art non-learning planners, showcasing its emergent coordinated behaviors and adaptability to more challenging scenarios and various team sizes, and finally deploy its learned policies on hardware in an aerial search task.

**Keywords:** Pursuit-Evasion, Graph Attention, Path Planning, MARL

## 1   Introduction

Pursuit-evasion broadly encompasses general problems that consider detecting/capturing mobile or even adversarial targets (i.e., *evaders*) by a team of *pursuer agents*. It has broad applicability in robotic deployments, including human localization after disasters, area patrols to prevent unauthorized activities, and environmental monitoring of wildlife habitats [1, 2, 3, 4]. In this work, unlike many pursuit-evasion problems that assume full observability of the evader's position and motion dynamics [5, 6, 7]—allowing the focus to be merely on chasing the evader(s) without the need for searching—we address a more general **worst-case scenario**: visibility-based pursuit-evasion, where agents have no information about the evaders. In this scenarios, agents with limited sensing range must coordinate to eventually detect all omniscient evaders, regardless of their movements.

In visibility-based pursuit-evasion, based on the agents' sequential observations, the environment is classified into *contaminated areas*, where the presence of evaders is uncertain, and *cleared areas*, where it is guaranteed that no evaders can reside/enter without being detected by any agent [2, 8, 9]. Agents are tasked with expanding *frontiers* (i.e., the boundaries between cleared and contaminated areas) to methodically explore the entire environment, turning all contaminated areas into cleared areas. During the task, agents need to coordinate their actions to control *recontamination*, where evaders might re-enter cleared areas undetected, turning them back into contaminated areas (see Fig. 1b). Therefore, this problem is also referred to as adversarial search [5], or the clearing problem [8, 10]. Due to the complexity of this problem, [8] deploys rule/constraint-based planners to reduce the complexity by prohibiting agents' actions that lead to recontamination, thereby significantly reducing the action space of agents. However, as shown in Fig. 1c, recontamination is unavoidable to complete the task in some challenging scenarios. Following works [10, 11] utilize centralized (meta-heuristic) sampling-based methods to seek efficient search directly in unconstrained

---

*Equal contribution. Correspondence to `wy98@u.nus.edu`

8th Conference on Robot Learning (CoRL 2024), Munich, Germany.

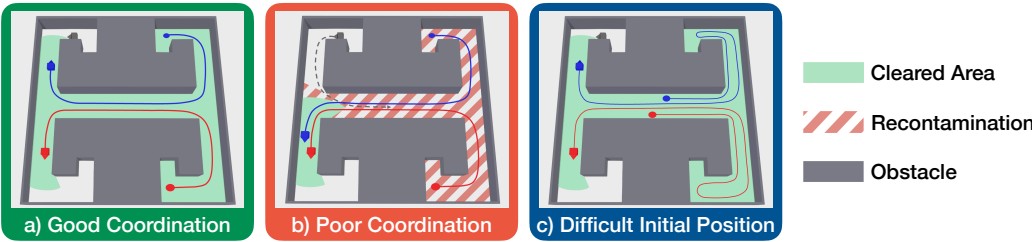

Figure 1: **Visibility-based pursuit-evasion in a simple environment with two agents.** The agents' trajectories are shown in red/blue, while that of an imaginary, worst-case evader is shown in gray.

action spaces. However, these methods are still limited to relatively small-scale scenarios or require extensive optimization.

Motivated by the advanced performance recently achieved by deep reinforcement learning (DRL) over sampling-based methods in various single-robot informative path planning tasks [12, 13, 14], we propose ViPER, a MARL-based solution for pursuit-evasion, allowing agents to learn to actively collaborate in a decentralized manner, to achieve significantly improved performance over conventional methods in relatively large-scale pursuit-evasion tasks. To develop such a desired learning-based planner, we make efforts to address the following key challenges: **(1)** Informatively yet concisely encode the observed/communicated information through the agents' neural network to enable efficient policy learning. Complex observations (including cleared/contaminated areas, obstacles, other agents, etc.) need to be represented in an efficient format so that the neural network can learn smoothly and efficiently. **(2)** Accurately model the long-term influence of agents' decisions on *recontamination*. Due to the changeability of the task (i.e., environments are partially known), model-free RL may struggle at accurately estimating the underlying transition model, making the training process less stable. **(3)** Efficiently enable agents to learn cooperative maneuvers. In some scenarios, the optimal actions of the team might depend on sensible decisions of specific agents, making efficient credit assignment critical for cooperation learning. ViPER addresses the challenges above by:

- incorporating the cleared area dynamics and the other agents' information into a graph attention network to allow context-aware policy learning;
- leveraging privileged ground-truth information during training to address the inaccuracy and instability inherent to model-free RL's state value estimation;
- employing an attentive critic for credit assignment that integrates agent action features to enable agents learning more efficient, decentralized cooperation.

To the best of our knowledge, ViPER is the first neural solution for the visibility-based pursuit-evasion problem that considers worst-case adversaries. Our approach improves team coordination, significantly outperforming state-of-the-art non-learning planners in unknown environments. We further empirically show that ViPER, once trained, can adapt to varying team sizes and (significantly more challenging) map structures. We also highlight its robustness in the face of agent failure, and finally experimentally validate learned policies on hardware in an aerial search task.

## 2 Prior Works

The initial formulation of the guaranteed worst-case pursuit-evasion problem was introduced by Parsons [15], followed by Suzuki and Yamashita [16] who proposed a visibility-based variant, transitioning the setting from discrete graphs to continuous polygonal environments. In such scenarios, agents aim to find an evader who can move at unpredictable and potentially unbounded speeds.

**Known environments:** There is a rich literature on visibility-based pursuit-evasion problems, with the majority of works considering fully known environments in which agents precalculate their routes offline before execution. Many of these studies focus on completeness and optimality of single-agent pursuit-evasion algorithms, considering an omnidirectional sensor with infinite

range [17, 18]. Furthermore, Gerkey et al. [2] provided a complete algorithm for agents with a limited angle of view and proved that computing the minimum number of such agents required to search an area is NP-hard. Since there is no known closed-form solution for the multi-agent pursuit-evasion game, Quattrini Li et al. [19] decomposed the environment into convex hulls and introduced a heuristic search approach, whereas Olsen et al. [11] proposed solving it with graph sampling-based method, aiming to mitigate the inherent complexity of the complete algorithms [20].

**Unknown environments:** Sachs and Rajko [21] presented the first single-agent algorithm that does not require a complete map of the environment, while Kolling and Carpin [9] utilized coordinated sweep lines by the team to simultaneously clear and explore unknown environments. Based on this, Durham et al. [8] developed an algorithm that guarantees capture if the number of pursuers is sufficient. They introduced a method to update the global frontiers between cleared and contaminated areas using solely local information. All these methods assume a relatively simple polygonal topology of the unknown environments and require more than sufficient agents to carry out the tasks [10].

## 3 Problem Formulation

**Environments:** We consider a bounded, multiply-connected environment represented by an occupancy map $\mathcal{M} \subset \mathbb{R}^2$ that is initially known ($\mathcal{M} = \mathcal{M}_k$) or unknown ($\mathcal{M} = \mathcal{M}_u$) to the agents. In the case of an unknown environment, the map is gradually explored and revealed by the agents such that $\mathcal{M}_k \cup \mathcal{M}_u = \mathcal{M}$. Based on their omnidirectional sensor measurements, agents will classify the known areas into either traversable free areas $\mathcal{M}_f$ or occupied areas $\mathcal{M}_o$ within their *line-of-sight* visibility range (i.e., $\mathcal{M}_f \cup \mathcal{M}_o = \mathcal{M}_k$). The free space is further categorized into either cleared areas $\mathcal{M}_c$ or contaminated areas $\mathcal{M}_f \setminus \mathcal{M}_c$. Note that we assume perfect communication among agents, allowing them to exchange information and share the same map throughout the task.

**Agents and evaders:** We task $n$ agents to coordinate their trajectories $\Psi = \{\psi_1, ..., \psi_n\}$, with the objective of capturing evaders capable of moving at potentially unbounded speeds $v_e$. Here, $\psi_i : \{0, 1, 2, ...\} \rightarrow \mathcal{M}_f$ represents the trajectory of agent $i$ over time, with $\psi_i(t)$ indicating the agent's location at timestep $t$. Agents scan their surrounding environment within their sensor footprint $S(\psi_i(t)) \subseteq \mathcal{M}$ which has a range of $r_{\text{fov}}$ (with visibility also limited by obstacles), communicate with other agents to exchange their measurements, and consequently update their map $\mathcal{M}$. We hypothesize a worst-case scenario where the evaders are positioned right outside the frontiers between the cleared and contaminated areas, denoted by $\mathcal{F} = \partial \mathcal{M}_c \setminus \mathcal{M}_o$, poised to intrude once there are any frontiers not within the visibility of any agent. Such frontiers are termed the *uncovered frontiers* $\mathcal{F}_u$, where evaders in contaminated areas can penetrate and cause recontamination, confined to their speed $v_e$. Since the locations of an agent in two consecutive timesteps are within each other's visibility (explained in Section 4), the intersection of two consecutive sensor footprints will also be visible during the agent's movement. This applies to any agent in the team and we denote this intersected area as

$$S(\Psi_{t:t+1}) = \bigcap_{t'=t}^{t+1} \left[ \bigcup_{i=1}^{n} S(\psi_i(t')) \right]. \tag{1}$$

Recontamination will spread through the free areas $\mathcal{M}_f$ at speed $v_e$, encroaching upon the cleared areas $\mathcal{M}_c$. In particular, when evaders are arbitrarily fast (i.e., $v_e = \infty$), the contaminated areas will expand through each connected $\mathcal{M}_f$ until reaching the boundary of $S(\Psi_{t:t+1})$.

**Objective:** The objective of the visibility-based pursuit-evasion problem is to determine the optimal agent trajectories $\Psi^*$ such that the cleared areas can expand to encompass the entire free space while minimizing the length of the trajectory for all agents:

$$\Psi^* = \arg\min_{\Psi} \sum_t \max_i \|\psi_i(t+1) - \psi_i(t)\|, \text{ s.t. } \mathcal{M}_c = \mathcal{M}_f^*, \tag{2}$$

with $\mathcal{M}_f^*$ the ground-truth free areas. Note that we optimize the sum of the maximum length per step, assuming synchronous decision-making and execution, whereby all agents must reach their viewpoints to proceed to the next step.

# 4 Multi-Agent Coordination Method

This section casts visibility-based pursuit-evasion as an RL problem, outlines our proposed policy and critic networks based on graph attention, and details its training procedure. We provide insights into the design of our observation, reward, and network architecture to foster advanced coordination.

## 4.1 Pursuit-Evasion as an RL Problem

**Sequential decision-making:** While agents can pre-plan their trajectories offline in a known environment, navigating an unknown environment relies on an adaptive online planner to continually update the map. This casts the task as a sequential decision-making problem, specifically a Markov decision process (MDP). We first uniformly distribute candidate viewpoints (also referred to as nodes hereafter) $V_t = \{v_0, v_1, ...\}$, $\forall v_j = (x_j, y_j) \in \mathcal{M}_f$ across the free areas, and append additional nodes into the newly identified free areas (if any) with each decision timestep. We then connect each node with its collision-free neighboring nodes within a specified range (usually $\leq r_{\text{fov}}$), forming the graph $G_t = (V_t, E_t)$ with its corresponding traversable edge set $E_t$. Upon all agents reaching their previously selected viewpoints, each agent will simultaneously choose the next viewpoint to visit, sequentially constructing the agent's trajectory $\psi_i(t) \in V_t$.

**Observation space:** Inspired by recent single-agent DRL-based informative path planning planners [12, 13], the observation $o_t = (G'_t, \Psi_t)$ includes the *augmented graph* $G'_t = (V'_t, E_t)$ and the current locations of the agent team $\Psi_t$. The augmented graph $G'_t$ shares the same edge topology $E_t$ as $G_t$, but also incorporates map information to assist agents in making informed and coordinated decisions. For each $v'_j \in V'$, the property of the node is $v'_j = (\Delta x_{i,j}, \Delta y_{i,j}, p_j, u_j, u_{u,j}, s_j, s_{c,j}, g_j)$: **(1)** Relative position $(\Delta x_{i,j}, \Delta y_{i,j})$: the node's position relative to agent $j$, i.e., $(\Delta x_{ij}, \Delta y_{ij}) = v_j - \psi_i(t)$. For simplicity, we omit the agent enumerator $i$ in graph and observation. **(2)** Occupancy of other agents $p_j$: a binary value indicating whether any other agent is positioned at $v_j$. **(3)** Utility $u_j$: represents the number of visible frontiers at node $v_j$ [22], i.e., $u_j = |\{f \in \mathcal{F} : \|f - v_j\| < r_{\text{fov}}, \overline{v_j f} \cap (\mathcal{M}_o \cup \mathcal{M}_u) = \emptyset\}|$, where $\overline{v_j f} \subset \mathcal{M}$ is a straight line connecting the node to the frontier. The node utility in contaminated areas is set to $-1$. **(4)** Uncovered utility $u_{u,j}$: similar to utility, but counts only the uncovered frontiers not within any agents' visibility, i.e., replace $\mathcal{F}$ above with $\mathcal{F}_u$. **(5)** Cleared signal $s_j$: a binary value indicating whether $v_j$ is within cleared areas. **(6)** Counterfactual cleared signal $s_{c,j}$: hallucinates a scenario without the presence of agents, considering how cleared areas would change, to help improve understanding of the evader's speed. Specifically, when $v_e = \infty$, $s_{c,j} \equiv 0$. **(7)** Guidepost $g_j$: a binary value indicating whether $v_j$ is on the agent's A$^*$ trajectory to its nearest frontier, which significantly aids in agent navigation to frontiers, especially when they are distant. The node coordinates and utility values in the observation are normalized before being fed into the neural network.

**Action space:** Each time all agents reach their designated viewpoints $\Psi_t$, the graph attention-based networks, parameterized by $\theta$ and shared among all agents, output stochastic policies $\pi_\theta(a_{i,t} \mid o_{i,t})$, $i = 1, ..., n$ (s.t. $\psi_i(t + 1) = v_j, (v_j, \psi_i(t)) \in E_t$). Agents select the next neighboring node based on these policies, then navigate to their selected viewpoints along straight paths, updating their maps with the information collected en route. If multiple agents select the same viewpoint, the others will alternatively choose the closest node to that viewpoint.

**Reward structure:** Our reward is designed to incentivize agents to optimize the task objective as outlined in Eq. (2). That is, we assign rewards/penalties for expanding/shrinking cleared areas, impose penalties for the largest distance traveled between viewpoints among all agents, and assign a substantial reward $r_f$ upon the successful task completion (0 otherwise):

$$r_t = c_1 \cdot (|\mathcal{M}_{c,t+1}| - |\mathcal{M}_{c,t}|) - c_2 \cdot \max_i \|\psi_i(t+1) - \psi_i(t)\| + \mathbb{1}_{\{\mathcal{M}_c = \mathcal{M}_f^*\}} \cdot r_f, \quad (3)$$

where $c_1, c_2$ are two constants for scaling. This reward $r_t$ is equally shared among all agents, as it is challenging to design a reward structure that aligns with the actual objective while accurately assigning individual rewards to each agent.

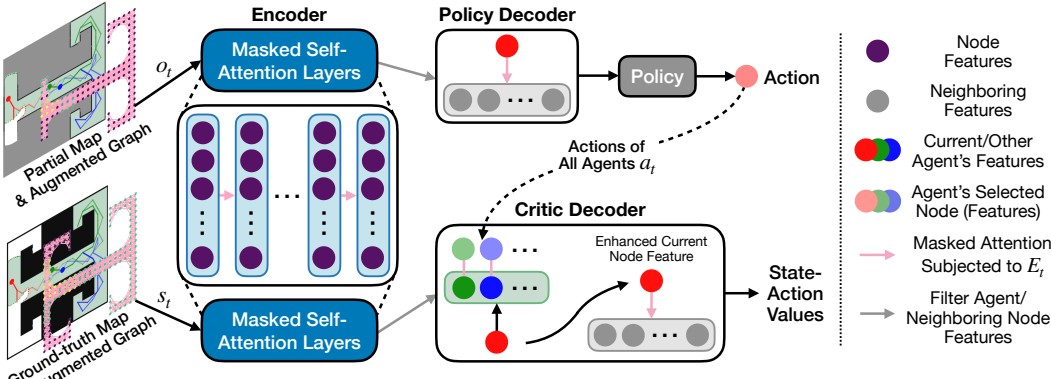

Figure 2: **ViPER's policy and critic networks based on graph attention.** It illustrates the scenario with unknown prior knowledge of the environment. In the constructed maps, black, white, and gray colors denote the occupied, free, and unknown areas, respectively. The collision-free graph is extracted from the map, with nodes colored to represent their utility and connected by pink edges. Note that the ground-truth map is only used by the critic during training.

## 4.2 Network Architecture

Inspired by recent advances in graph attention networks [23, 24], we design a network to learn policy $\pi_\theta$ that orchestrates coordinated behaviors among multiple agents. Our graph attention-based policy network consists of an encoder and a decoder: the encoder extracts information from nodes across the entire known map, while the decoder yields which neighboring node to visit next, based on the aggregated global information.

We train a critic network to approximate the state-action values. These values represent the expected cumulative future rewards (i.e., return) and guide our policy network by evaluating the quality of the chosen actions. The environment is fully observable if it is known a priori: $s_t = o_t$. However, in unknown environments, the policy network can only access partial information about the environment as agents explore. In our work, unlike the policy network, the critic network is provided with complete ground-truth information (e.g., entire map/graph, see Fig. 2) during training phase. This *privileged learning* approach, leveraging the critic's access to the full state, contributes to a more stable training process [13]. To harness the full potential of the well-established centralized training with decentralized execution (CTDE) paradigm [25], we further provide the critic with the actions of other agents within the team $a_t$. That is, during the training phase, the integration of the state $s_t = (G_t^{'*}, \Psi_t)$ along with $a_t$ enables the critic to estimate state-action values with greater accuracy, thus in turn implicitly assisting agents in predicting the structure of the environment as well as the behaviors of other agents.

**Encoder:** Our encoder is essentially a sequence of masked self-attention layers (introduced in Appendix A) that enhances the features of each node by incorporating information from other nodes within the graph structure. We first apply a feed-forward layer to embed the node properties in $V_t'$ into $d$-dimensional node features $h^n$. These node features are then processed through $N$ stacked self-attention layers (i.e., $h^q = h^{k,v} = h^n$, and with $N = 6$ in our practice), where each layer uses the output of the previous one as its input. We calculate the mask $M_t$ of the graph using the adjacency matrix derived from $E_t$. This mask is applied to ensure each node has access only to features of its neighboring nodes (i.e., attention weight $w_{i,j} = 0$, $\forall (v_i, v_j) \notin E_t$). Although attention is constrained to neighboring nodes at each layer, nodes can still acquire non-neighboring node features by repeatedly aggregating features through this stacked self-attention structure. We find that, in contrast to unmasked self-attention, our method of masked self-attention, which propagates neighboring features layer by layer, leads to a more effective learning of pathfinding solution [26]. The encoder thus enhances the node features $h^n$ to $\tilde{h}^n$, as each updated set of node features contains dependencies on other nodes in the graph, depending on $M_t$ and $N$.

**Policy decoder:** The decoder is used to determine the final policy based on the enhanced node features $\tilde{h}^n$. From these node features, we select the one at the robot's current position as query:

$h^q = h^c = \tilde{h}^n_{\psi_i(t)}$. The current node features $h^c$, along with its neighboring node features (presented as key-value pairs) from $\tilde{h}^n$, are fed into an attention layer, thereby enhancing the output $\tilde{h}^c$ with information from its neighboring nodes. Subsequently, we adopt a single-head attention mechanism with $\tilde{h}^c$ and its neighboring node features, and directly use its attention scores as the final output policy $\pi_\theta = (a_{i,t} \mid o_{i,t}) = w_{i,j}$, from which the agent selects the next viewpoint to move to. This design relaxes the need for a predefined policy size, instead allowing the policy's dimensions to adapt dynamically to the number of neighboring nodes, thus naturally accommodating graphs with arbitrary connectivity.

**Critic decoder:** The critic network $Q_\phi$, parameterized by $\phi$, shares the same encoding structure with the policy network $\pi_\theta$, but employs a different decoder that incorporates other agent's actions. We mirror the idea in MAAC [27] that introduced an attentive critic to select relevant information for each agent. From the enhanced node features $\tilde{h}^n$ output by the encoder, we filter the node features of other agents $\tilde{h}^n_{\psi_{-i}(t)}$ and their corresponding selected features $\tilde{h}^n_{\psi_{-i}(t+1)}$ at the next timestep based on their current actions $a_t$. These features are then concatenated as key-value pairs to enhance the current node feature $h^c$. Subsequently, informed by the actions of other agents, the current node features are further enhanced by incorporating neighboring features, identical to the process in the policy decoder. The enhanced current node features $\tilde{h}^c$ are then concatenated with neighboring features and are finally mapped to state-action values $Q_\phi(s_t, a_t)$ by a feed-forward layer.

### 4.3 Network Training Procedure

We train our RL agents using the soft actor-critic (SAC) algorithm on $4,000$ randomly created maps for our training dataset, and an additional set of 100 unseen maps as the test dataset. The training details (algorithm, dataset, graph and sensing configurations) can be found in Appendix B. ViPER's code and trained model are available at https://github.com/marmotlab/ViPER.

## 5 Experiments

In this section, we compare our method with state-of-the-art (SOTA) baseline planners designed for unknown environments, showcasing its superiority in terms of performance, generalization, and robustness. We also discuss the emergent coordinated behavior of our trained policy and present the results of a real-life aerial search task involving three drones and one ground robot.

### 5.1 Comparison Analysis

We compare ViPER with two planners from Durham et al. [8] and Gregorin et al. [10]. Durham et al. proposed a distributed planner that makes decisions based solely on local frontiers and communication, thus it does not require prior mapping of the environment. Suggested by the settings in their original paper, we tuned their planner to output high-quality solutions for a fair comparison. While many studies evaluate their algorithms in simply structured environments, our map dataset is more structurally complex, requiring agents to implicitly predict the map and actively respond to updates. Our evaluation map dataset comprises 100 maps that were unseen during model training. We also use multiply-connected and conjugated maps from Gregorin et al. and compare our performance against the results reported in their paper, using the same $r_{\text{fov}}$. Following [10], we test each map 30 times with random start positions. Note that throughout all evaluations, we use our model trained with 3 agents and a bounded evader speed ($v_e \leq 20\text{m/timestep}$, with map size $100 \times 100\text{m}^2$) to facilitate faster environment simulation during training.

We report the success rate and the maximum trajectory length $\sum_{t=0}^{T} \max_{i=1}^n \|\psi_i(t+1) - \psi_i(t)\|$ in Table 1, averaged across 100 maps with unbounded evader speed, i.e., $v_e = \infty$. We mark failure of the task if the trajectory length of any agent exceeds $1,000\text{m}$. Our results show that ViPER outperforms the other two baselines in both metrics. Upon closer inspection, the performance gap is more pronounced when the team size is smaller. We believe that this likely comes from the restriction in Durham's algorithm of not allowing recontamination, potentially requiring a larger team size than

Table 1: **Comparison against SOTA planners in different map sets and team sizes** $n$**.** We report the success rate and averaged trajectory length (in parentheses).

| | **Our maps** | | | **Multiply-connected maps [10]** | | | | **Conjugated maps [10]** | | |
|---|---|---|---|---|---|---|---|---|---|---|
| $n$ | Durham | ViPER | $n$ | Gregorin | Durham | ViPER | $n$ | Gregorin | Durham | ViPER |
| 3 | 24.0% (842.7) | **86.0%** (411.5) | 4 | 38.5% | 20.0% | **50.0%** | 9 | 29.5% | 84.2% | **92.5%** |
| 4 | 66.0% (577.1) | **100%** (281.9) | 5 | 73.5% | 60.0% | **80.0%** | 15 | 48.5% | 88.3% | **100%** |
| 5 | 89.0% (394.4) | **100%** (230.0) | 6 | 74.0% | 76.7% | **98.3%** | | | | |

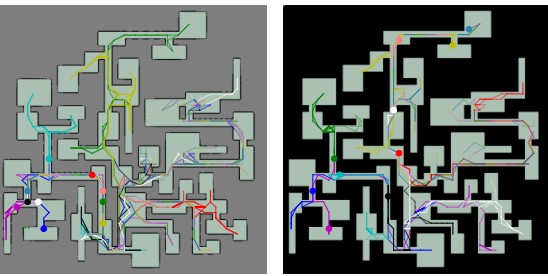

Figure 3: **Paths in complex unknown/known (left/right) large-scale environments** ($n = 10$)**.**

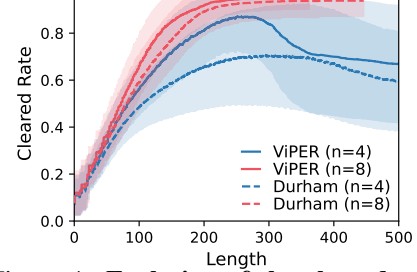

Figure 4: **Evolution of the cleared rate versus trajectory length on our map set.**

necessary. This is evident in Fig. 4, which plots the evolution of the cleared rate $|\mathcal{M}_c|/|\mathcal{M}_f^*|$ compared to trajectory length. When $n = 4$, the performance disparity is significant, while at $n = 8$, both approaches achieve a 100% success rate, resulting in averagely monotonic increases in the cleared rate. Gregorin's approach, on the other hand, is centralized and allows a certain level of recontamination. Their method combines random walking with an evolutionary approach to address rarely explored states. However, in conjugated maps, its success rate still significantly drops as the environment size and the number of agents increase, limiting its scalability to larger team sizes.

## 5.2   Generalization and Adaptability Analysis

**Known environments:** We evaluate our model in known environments and present our results in Table 2. Despite being trained in unknown environments, our model is capable of adapting to known environments (and vice versa) without much performance change/degradation, and it scales effectively to various team sizes. This is largely due to the fact that the agent observations only include the number of observable frontiers between cleared and contaminated areas (i.e., utility) by design, while excluding information on the *exploration frontiers* between known and unknown areas. That is, we regard the exploration of the environment as an auxiliary task of the pursuit-evasion.

**Complex environments:** We adopt an additional map dataset of 100 maps with significantly more complex topologies than those used during training and 4 times larger the area to evaluate the algorithm's generalizability with 10 agents. There, the task is deemed unsuccessful once any agent's trajectory length exceeds $3,000$m. The results are presented in the last row of Table 2 and example solutions can be seen in Fig. 3. In unknown environments, the agents may initially misjudge the layout, leading to inaccurate agents distribution. Consequently, certain agents need to wait at junctions for a long period, awaiting the arrival of other agents before continuing to collaboratively explore the environment and minimize recontamination, thereby increasing the difficulty of the task. We believe these results suggest that ViPER can generalize to vastly different/challenging environments with minimal impact on map size or the number of agents involved, without the need for retraining.

**Agent failure:** In Appendix C, we exhibit ViPER's robustness to mid-task agent failures, showing its ability to adapt and prevent further recontamination from failed agent(s).

## 5.3   Emergent Coordination

Our agents exhibit interesting emergent coordinated behaviors during training, thanks to our graph-attention-based network and centralized privileged critic (an ablation study of these modules can be found in Appendix D). One of our map set is shown in Fig. 5, where we display the solution

Table 2: **Adaptability to various environments/settings.**

| $n$ | Our maps | |
| --- | --- | --- |
| | Unknown | Known |
| 3 | **86.0%** (411.5) | 82.0% (401.3) |
| 4 | **100%** (281.9) | **100%** (267.6) |
| 5 | **100%** (230.0) | **100%** (230.7) |
| | **Our maps (complex)** | |
| 10 | 69.0% (1795) | **98.0%** (1005) |

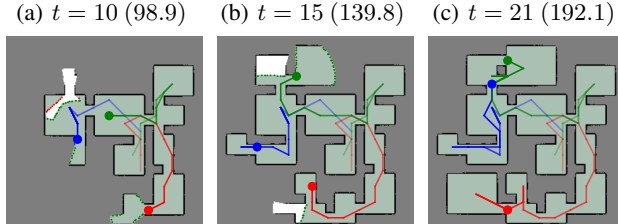

(a) $t = 10$ (98.9)    (b) $t = 15$ (139.8)    (c) $t = 21$ (192.1)

Figure 5: **3 agents clearing an unknown environment.** Dotted green/red lines show covered/uncovered frontiers.

snapshots at various timesteps and trajectory lengths (in parentheses). We observe that, once the green agent clears the top-right room and begins moving towards the blue agent for assistance, the blue agent starts to move downward in advance upon observing the green agent's approach, allowing for some areas of controllable recontamination to reduce trajectory length. Then, after clearing the room, the blue agent joins forces with the green agent to clear the upper room, which has an obstacle situated in the middle requiring at least two agents to clear. Our additional examples in Appendix E further confirm that, although agents make synchronous decisions in MARL framework, they can intelligently allocate themselves on-demand and dynamically redistribute as the task progresses.

## 5.4 Experimental Validation

We carried out experiments to validate ViPER in an aerial search task using three Crazyfile 2.1 drones as searching agents and one TurtleBot3 robot as an imaginary evader (see Fig. 6). The experiment was conducted in a $4 \times 4\text{m}^2$ mockup arena with a waypoint node resolution of $0.2\text{m}$. We opted to use the optical flow sensors on the drones for position tracking, with the origins set at their take-off positions, and transformed the updated positions into a global

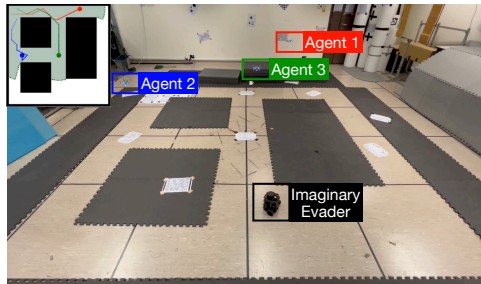

Figure 6: **Experimental validation on three Crazyfile drones.**

frame. Waypoints were iteratively published to drones by a remote computer, while the TurtleBot3, controlled by a human as an omniscient evader, only navigated in contaminated areas. This experiment confirms that ViPER can be deployed on robots with relatively low computational costs, and exhibit high performance. For each agent, forward inference takes less than $0.01\text{s}$, and the combined environment update and planning take less than $1\text{s}$ of wall clock time.

## 6 Conclusion

**Summary:** We introduce the first neural framework based on multi-agent reinforcement learning for visibility-based pursuit-evasion tasks that addresses worst-case scenarios. Our approach relies on a graph attention network to allow agents to individually process their shared information and achieve the necessary level of tight coordination that can ensure the detection of omniscient and potentially arbitrarily fast evaders. In our evaluation, our learned policies exhibit emergent coordinated teamwork behaviors that outperform state-of-the-art planners by a large margin. We also demonstrate ViPER can adapt to different team sizes, handle diverse scenarios, and remain effective in the face of agent failures.

**Limitations and future work:** Although ViPER can efficiently handle exploration and recontamination compared to other methods, the initial positions of the agents still greatly affects their performance and can sometimes result in task failure. When all agents are in a position that necessitates additional support from other agents, they are unlikely to sacrifice their own, usually extensive, cleared areas to assist others. This reluctance can lead to deadlocks in actions, despite the theoretical existence of a solution. Future work will explore multi-level graph representations and hierarchical decision-making for improved graph structure reasoning and state exploration.

**Acknowledgments**

This work was supported by Temasek Laboratories (TL@NUS) under grant TL/FS/2022/01.

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

# A    Attention Layers

Our network relies on attention layers in the Transformer [28], where each layer transforms a query vector $h^q$ and a collection of key-value pair vectors $h^{k,v}$ into an output vector $h'_i$ by computing a weighted sum of values. The attention scores are derived by measuring the similarities between queries and keys within each attention head:

$$w_{ij} = \text{Softmax}\left(\frac{q_i^\top \cdot k_j}{\sqrt{d}}\right), \; h'_i = \sum_{j=1}^n w_{ij}v_j, \qquad (4)$$

where $q, k, v$ are derived by three separate learnable matrices in $\mathbb{R}^{d \times d}$ as $q_i = W^Q h_i^q$, $k_i = W^K h_i^{k,v}$, $v_i = W^V h_i^{k,v}$. Following [29], we adopt Identity Map Reordering (IMR) between attention layers, placing layer normalization before attention layer to improve network stability in RL tasks.

# B    Network Training Procedure

**Algorithm:** We train our DRL agents with the soft actor-critic (SAC) algorithm. SAC learns a soft value function by modifying the policy gradient to include an entropy term, $\mathcal{H}(\pi(\cdot \mid o_t))$, which encourages the exploration of more diverse actions and helps prevent premature convergence [30]:

$$\nabla_\theta J_\pi(\theta) = \mathbb{E}_{o \sim D, a \sim \pi}[\nabla_\theta \log(\pi_\theta(a_{i,t} \mid o_{i,t})) \cdot (-\alpha \log(\pi_\theta(a_{i,t} \mid o_{i,t})) + Q_\phi(s_t, a_t))], \qquad (5)$$

where $T$ is the number of timesteps and $\gamma \in [0, 1]$ the discount rate, $D$ the replay buffer, and $\alpha$ serves as a temperature parameter that scales the importance of the entropy term against the reward. The temperature parameter is automatically adjusted to match the target entropy $\overline{\mathcal{H}}$.

**Map dataset:** We use a dungeon map generator [31] to randomly create a set of $4 \times 1,000$ multiply-connected maps for our training dataset, with each map oriented four times for map set augmentation, and an extra set of 100 different maps that are unseen during training for performance evaluation. Each map have a size of $100\text{m} \times 100\text{m}$.

**Sensing:** The onboard omnidirectional sensor with a radius of $r_{\text{fov}} = 20\text{m}$ will monitor/identify potential evaders (and map the environment). Here, assuming a circular field of view (FoV) aligns with most related works [9, 10] and helps incorporate ViPER with both downward-facing cameras and LiDAR. Each agent scans its surrounding environment and communicates with other agents to update a shared belief.

**Graph:** The candidate viewpoints $V_t$ are uniformly distributed throughout the entire known free areas $\mathcal{M}_f$. The spacing between two adjacent nodes (i.e., resolution) is set at $4\text{m}$. Note that these parameters are tunable to meet specific practical requirements. Each node is connected with its $k = 25$ nearest neighboring nodes (including itself) to establish the edge set $E_t$, provided the paths between them are collision-free. We consider a successful task completion to be the closure of the cleared areas $\mathcal{M}_c$ (i.e., no frontiers between cleared and contaminated areas).

**Training:** During training, the episode is terminated once its length exceeds 128 timesteps, with a replay buffer size of $|D| = 10,000$, a discount rate of $\gamma = 1$, and the target entropy set to $0.01 \cdot \log k$. Our model is trained on a workstation with an AMD 7950X3D CPU where 12 environments run in parallel, and an NVIDIA GeForce RTX 3090 GPU to train the network once an episode terminates, after accumulating $5,000$ steps of data in the replay buffer. The model converges after 40 hours of training, and we adopt the model at $40,000$ episodes for experiments in Section 5.

# C    Robustness to Agent Failure

We evaluate the robustness of ViPER to agent failures occurring in the middle of the task. Specifically, we simulate agent failure by deactivating a random agent, including its movement and sensor, once agents have explored half of the environment. The result is plotted in Fig. 7, averaged over 100

maps. Our results indicate that the performance remains similar to (or slightly superior to) that of the fully functional agent group (comparing $n \to n-1$ to $n-1$). This demonstrates that the remaining functional agents are able to reactively adapt and prevent further recontamination from the failed agent, thereby avoiding the need to almost entirely restart the task should the recontamination spread uncontrolled.

## D    Ablation Results

We perform a comprehensive ablation study to evaluate the effectiveness of our graph attention architecture, attentive critic, and privileged learning module.

**Graph attention encoding:** To evaluate the effectiveness of our graph attention architecture, we additionally train two ablation baselines. In comparison to ViPER with a 6-layer graph encoder, one baseline was trained with a 3-layer graph encoder, while the other was trained without any encoder. As shown in Fig. 8, the performance significantly degrades with a shallower encoder during training, as agents' limited graph perception results in more myopic decision-

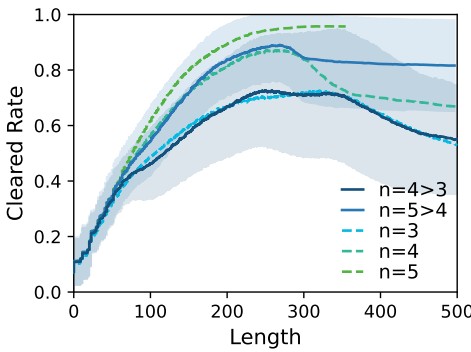

Figure 7: **Performance amidst agent failure during the task.**

making. Without a graph encoder, the agents struggle to learn any effective long-term policies necessary to successfully complete the task, leading to frequent task failures.

**Critic module:** We train three additional ablation baselines: ViPER without the MAAC (attentive critic module), ViPER without GT (ground truth information during training), and ViPER without both MAAC and GT. We present the reward and length curves during training in Fig. 8. It is evident that ViPER consistently outperforms its baselines throughout the training process. We further observe a significant performance decline without privileged learning, indicating the importance of access to ground truth information for the critic to achieve a more accurate value estimate during training. Our ablation study demonstrates that both the attentive critic and privileged learning (individually and combined) enhance ViPER's ability to learn a better policy for multi-agent coordination in visibility-based pursuit-evasion tasks.

## E    Emergent Multi-Agent Coordination: Some Examples

We handcraft several specific scenarios to showcase team coordination and provide insights into ViPER's policy. As illustrated in Fig. 9a, when there is an isolated obstacle/chamber in the center of a closed environment, it is impossible for a single agent to clear the entire areas, as the evaders can continuously hide behind the obstacle to elude detection. In this case, two agents must simultaneously extend their frontiers from both sides of the obstacle to ensure the evaders are detected. Similarly, as shown in Fig. 9b, each passage must be monitored by an agent to prevent recontamination. In Fig. 9c, four agents are initially paired off to direct toward rightward and downward. However, upon discovering more frontiers below, the green agent reactively reverses and adjusts its trajectory to head downward instead. Fig. 9d depicts scenarios where, if an agent inspects each room along a corridor, an evader could potentially slip past and conceal itself in an already cleared room. One solution is to use two agents, with one consistently monitoring the corridor. Our two agents alternatively choose to take turns clearing rooms and monitoring the corridor, resulting in an even shorter trajectory length.

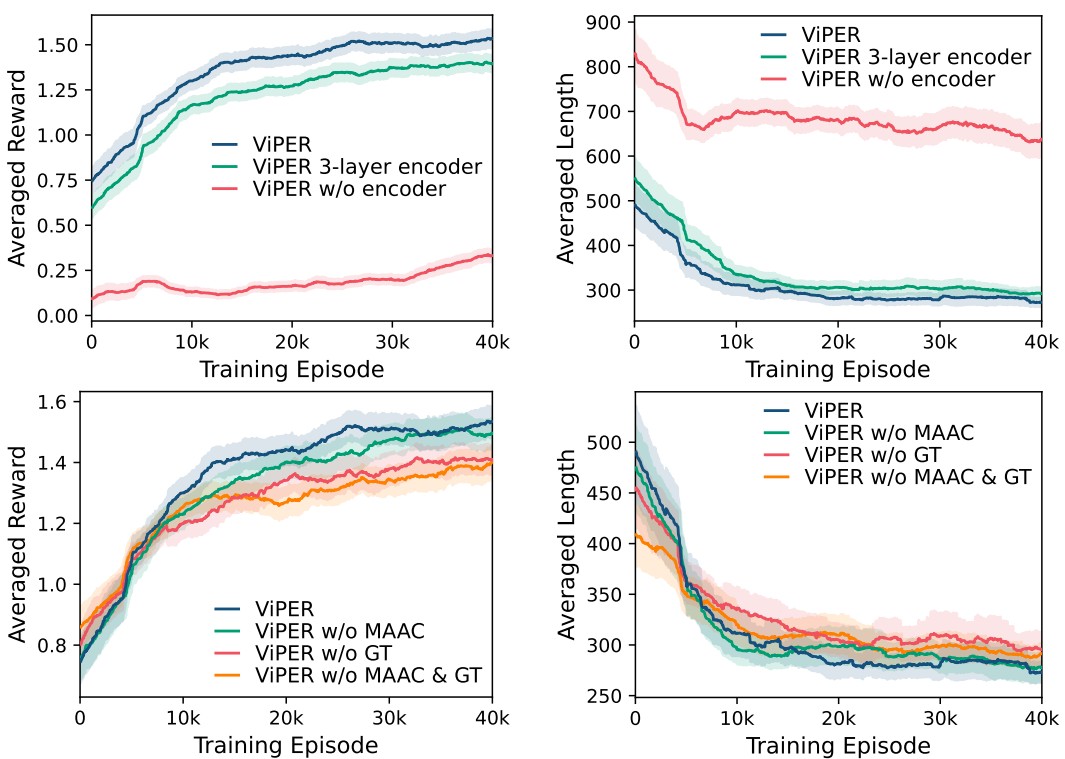

Figure 8: **Training curves of ViPER and its ablation baselines, all trained with a team size of** $n = 4$**.** All curves are averaged over a window of 200 data points, with the shaded area representing $\pm 0.2$ times the standard deviation.

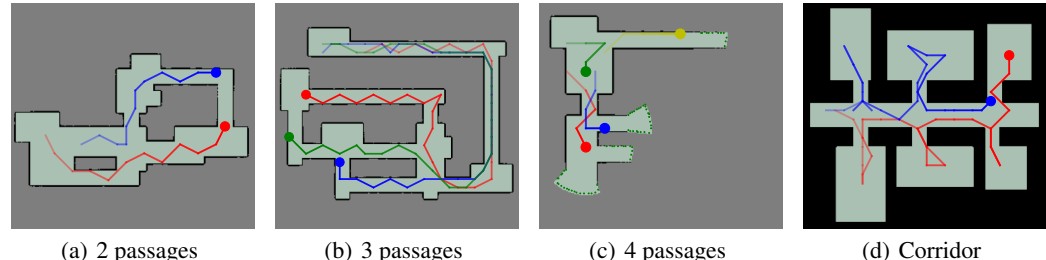

(a) 2 passages        (b) 3 passages        (c) 4 passages        (d) Corridor

Figure 9: **Coordinated behaviors among agents to clear the environment.** (a-c) unknown, (d) known.

