# OpenReview forum: "ViPER: Visibility-based Pursuit-Evasion via Reinforcement Learning"
_robot-learning.org/CoRL/2024/Conference — CoRL 2024_

### Official Review · Reviewer_ycfG · 2024-07-20

**Originality:** 3
**Technical Quality:** 3
**Clarity Of Presentation:** 4
**Potential Impact:** 3
**Recommendation:** 3
**Confidence:** 2

**Review:**

**Strengths**
- The masked graph attention network that extracts information across known nodes in order to coordinate future actions across the team appears is an intuitive approach to solving the pursuit-evader problem involving multiple agents.
- The method outperforms the reported baselines by a significant margin while accounting for recontamination
- The authors validated their approach with an albeit small team of aerial vehicles and a single evader.
- The method is capable of adapting to unknown environments thanks to the privileged critic learning / partial observability of any individual agent - training paradigm

**Weaknesses**
- I understand that real robot experiments are difficult and time consuming, but it would have been nice to see results involving more than three drones and a single evader, especially since the evaluation results included larger teams and maps.
- The authors claim to compare to SOTA planners, but the two baselines are papers from 2012 and 2017. While I do not directly work in this field, I would be incredibly surprised if there weren't other more relevant baselines. Further, the reported baselines are not data driven approaches. It would only be fair to see how this method stacks up against other similar learning-based methods.

**Quality Of The Limitations Section:**

3

**Questions For Rebuttal:**

1. How does your method scale / perform with multiple evaders?

2. Since the graph attention network incorporates other environment information alongside the graph, is it possible for this method to work with heterogenous robot teams? If not, what changes would be needed to enable this?

**Robotics Focus:**

3

**Summary Of Paper:**

This paper introduces a method that utilizes graph attention networks to learn centralized planning decentralized control policies for the pursuit-evasion task.

**Summary Of Recommendation:**

The authors put forward a strong learning-based method that is trained on a dataset of thousands of maps, leading to generalization and adaptability to unseen environments. The method was validated with real robot experiments. However, the robot experiments were relatively small in scale and the baseline comparisons seem weak.

---

### Official Review · Reviewer_Z6Zd · 2024-07-21
**Overall interesting paper solving a limited sensor range visibility pursuit evasion problem with multiple robots**

**Originality:** 3
**Technical Quality:** 3
**Clarity Of Presentation:** 4
**Potential Impact:** 3
**Recommendation:** 3
**Confidence:** 5

**Review:**

The paper addresses an interesting problem in the pursuit-evasion space and approaches it with a technically sound learning-based approach. It is interesting to see it formulated in this way to have a practical way to solve the problem, rather than having only theoretical analysis and having an algorithm that addresses the worst case. The paper appears to provide convincing results with experiments performed in unseen environments, as well as with real robots.

It would be useful to perform an ablation study to actually see the impact of the graph-attention based network as claimed in the results.

Specific clarifications are needed:
- it would be useful to discuss also the failure cases for the proposed method. Is it happening for the fact that there are not enough agents, as discussed for the other methods? Or is there something else happening?
- it seems that the problem is to detect the evaders based on the objective, however, the paper also mentions about the capture of them in the problem formulation. Please clarify it.
- with the average results it is useful also to report the standard deviation.
- one limitation worth including is also the guarantee that the proposed method can provide.

Presentation/typos comments:
- Following works [10, 11] utilize -> Some works [10, 11] utilize

**Quality Of The Limitations Section:**

3

**Questions For Rebuttal:**

1. It would be useful to include an ablation study on the proposed architecture to see the impact of each component.

**Robotics Focus:**

4

**Summary Of Paper:**

The paper proposes a method for finding a plan for multiple robots with limited sensor range to clear an environment so that all potential evaders are found. The problem is solved formulating it as a multi agent reinforcement learning problem, with the method that is based on a graph attention network to learn the distributed policy. Experiments in simulation and one demonstration with real robots are performed.

**Summary Of Recommendation:**

While there can be an additional ablation study, overall the paper proposed an interesting learning-based method to solve a pursuit evasion problem, with convincing results, including experiments in unseen environments.

---

### Official Review · Reviewer_fEXc · 2024-08-01
**Simulation based research, limited real world robotics evaluation**

**Originality:** 3
**Technical Quality:** 3
**Clarity Of Presentation:** 3
**Potential Impact:** 2
**Recommendation:** 3
**Confidence:** 3

**Review:**

The paper proposes a neural solution that leverages a graph attention network to learn a coordinated
yet distributed policy via multi-agent reinforcement learning, verifying the idea with the pursuit-evasion task.
However, as far as this reviewer comes from a robotics control background, the motivation for such a methodology is not grounded in robotics, and the application of pursuit-evasion appears to be a toy problem.
The presentation of the originality and the contributions can be enhanced. Especially, considering what kind of variables are inputted to the model, and what kind of variables are outputted to control the robotic systems.
The authors claim that the problem is complex, however, the description is abstract and it is hard for readers from a robotics background to acknowledge the difficulty.
Strength:
* The theoretical background is well presented with mathematical correctness.

Weakness:
* Less graphical explanations, such as system diagram, how aerial robotic systems integrate with the model, what the variables controlled, and how the perception is performed.

**Quality Of The Limitations Section:**

2

**Questions For Rebuttal:**

The paper appears well-written and could potentially be enhanced by addressing the following points:

* The motivation for targeting the pursuit-evasion problem and the proposed approach should be clarified to improve readability, particularly for those primarily familiar with robotics.
* In the experimental evaluation, it should be specified whether the observed camera images are from the aerial robot or if they capture the entire field.
* The method for tracking the position of the drones should be detailed. Specifically, if coordinate transformations are used, what is the origin of these coordinates?
* The actions outputted by the model should be defined. Are they the desired velocity of the drone or the path of the drone?
* The definition of visibility during the experiments should be clarified. Is the drone equipped with a camera that scans the environment?

These clarifications will enhance the understanding and applicability of the research findings.

**Robotics Focus:**

2

**Summary Of Paper:**

This paper introduces ViPER, a new approach using a graph attention network and multi-agent reinforcement learning to improve coordination in visibility-based pursuit-evasion tasks where robots detect evaders in complex environments. ViPER outperforms traditional methods by adapting to various scenarios and team sizes, and has been successfully deployed on aerial search hardware.

**Summary Of Recommendation:**

The research is presented mainly from a theoretical perspective, and the real-world experiments appear to be limited. Therefore, from the viewpoint of robotics, including applications, the research could still be improved.

---

### Author Rebuttal · Authors · 2024-08-14

To evaluate the effectiveness of our graph attention architecture and address the concern raised by the Reviewer Z6Zd, we conducted an additional ablation study by varying the number of layers in our graph encoder. The results, presented in the new supplementary material, demonstrate that a shallower encoder leads to performance degradation, and without an encoder, agents can hardly learn any useful policies.

---

### Decision · Program_Chairs · 2024-09-04

**Decision:**

Accept

**Comment:**

Strengths:

• Novel approach using graph attention networks and multi-agent reinforcement learning for pursuit-evasion tasks
• Outperforms state-of-the-art non-learning planners by a significant margin
• Demonstrates adaptability to unknown environments and various team sizes
• Successfully deployed on real aerial robots, validating practical applicability

Weaknesses:

• Insufficient graphical explanations of system architecture and integration
• Limited real-world experiments (only 3 drones and 1 evader tested)
• Comparison only to older baselines (2012, 2017), missing more recent learning-based approaches